# The Use of Yoga to Manage Stress and Burnout in Healthcare Workers: A Systematic Review

**DOI:** 10.3390/jcm8030284

**Published:** 2019-02-26

**Authors:** Rosario Andrea Cocchiara, Margherita Peruzzo, Alice Mannocci, Livia Ottolenghi, Paolo Villari, Antonella Polimeni, Fabrizio Guerra, Giuseppe La Torre

**Affiliations:** 1Department of Public Health and Infectious Diseases, Sapienza University, 00185 Rome, Italy; rosario.cocchiara@uniroma1.it (R.A.C.); alice.mannocci@uniroma1.it (A.M.); paolo.villari@uniroma1.it (P.V.); 2Department of Oral and Maxillofacial Sciences, Sapienza University, 00161 Rome, Italy; margherita.peruzzo@gmail.com (M.P.); livia.ottolenghi@uniroma1.it (L.O.); antonella.polimeni@uniroma1.it (A.P.); fabrizio.guerra@uniroma1.it (F.G.)

**Keywords:** healthcare workers, stress management, burnout, yoga, occupational medicine, occupational health

## Abstract

The purpose of this systematic review is to analyze and summarize the current knowledge regarding the use of yoga to manage and prevent stress and burnout in healthcare workers. In February 2017, a literature search was conducted using the databases Medline (PubMed) and Scopus. Studies that addressed this topic were included. Eleven articles met the inclusion criteria. Seven studies were clinical trials that analyzed yoga interventions and evaluated effectiveness by gauging stress levels, sleep quality and quality of life. A study on Chinese nurses showed statistical improvement in stress levels following a six-month yoga program (χ2 = 16.449; *p* < 0.001). A population of medical students showed improvement in self-regulation values after an 11-week yoga program (from 3.49 to 3.58; *p* = 0.04) and in self-compassion values (from 2.88 to 3.25; *p* = 0.04). Four of the included articles were observational studies: They described the factors that cause stress in the work environment and highlighted that healthcare workers believe it is possible to benefit from improved physical, emotional and mental health related to yoga activity. According to the literature, yoga appears to be effective in the management of stress in healthcare workers, but it is necessary to implement methodologically relevant studies to attribute significance to such evidence.

## 1. Introduction

Healthcare professionals daily face heavy stress loads. Even during training, medical students and healthcare professionals must cope with heavy workloads and high stress levels in a competitive environment that facilitates the onset of burnout [1,2]. 

This pattern continues to be equally complex in the working environment where professionals must cope with a demanding job, the stress of shiftwork and various responsibilities derived from their chosen medical field [3].

The term “burnout” is usually referred to helping professions. It indicates a prolonged exposure to physical and psychological stress that results in deterioration of the emotional functioning of the individual and in a difficulty of adaptation to their professional commitments [4].

Most of the published literature regarding this topic focuses on the nursing category as it represents the most prominent group of healthcare providers and effectively combines the main health problems: Both psychological and physical stress [5,6,7]. Other professionals, such as surgeons and dentists have exhibited similar issues: These must be added to the numerous working hours they spend in stationary and incorrect posture from which musculoskeletal conditions such as back pain, neck or shoulder strain, headache and carpal tunnel syndrome derivate [8].

This pattern has captured the attention of healthcare managers who are now seeking solutions to these problems [9]. Burnout and stress impact the physical and mental health of the individual and compromise the quality of care as they generate dissatisfaction, anxiety and an increased risk for patients [10,11,12].

In consideration of these premises, the published literature indicates that yoga and other mind-body meditation programs provide some innovative solutions, scientifically recognized as effective methods to enhance empathy, reduce stress and improve physical work-related issues in healthcare professionals [13,14].

In some cases, yoga programs have been readapted to meet the needs of the work environment as being structured in less time-invasive sessions to be held in the workplace, in association with daily meditation to be carried out individually at home [15]. The clinical trials have highlighted significant efficacy in stress management, the reduction of burnout and in overall improvement of quality of life (QoL).

The aim of this study is to analyze and summarize all the current knowledge concerning yoga as an effective technique for the prevention and management of stress and burnout among healthcare workers.

## 2. Materials and Methods

### 2.1. Identification of Relevant Studies

This systematic review was performed following PRISMA (Preferred Reporting Items for Systematic Reviews and Meta-Analyses) criteria [16]. Two electronic databases were examined: Medline (PubMed) and Scopus. The following search algorithm was applied: “yoga AND stress management AND health professional”. In February 2017, eligible studies were selected through a multi-step approach (title reading, abstract and full-text assessment) by two independent researchers.

### 2.2. Study Selection and Eligibility Criteria

Search results found in both databases (PubMed and Scopus) were uploaded for screening into JabRef 3.8.1. A first selection was performed by filtering duplicates and subsequently, a title and abstract screening was conducted. All potentially relevant articles were then independently reviewed for full text and assessed for eligibility. Studies were included if they addressed the question of yoga used to manage stress and reduce physical pathologies in healthcare professionals. No restriction of year of publication or language was applied during the study selection. Any disagreement between the reviewers was resolved through a consensus session with a third reviewer. Figure 1 shows the flowchart of the selection of articles.

### 2.3. Data Extraction

The same selection strategy was used for data extraction: two different reviewers collected the data and any disagreement was resolved by a consensus session. The following information were gathered: Author; study design (observational study or clinical trial); year of publication; country; outcomes; quality score. After analyzing the full-text, the main identified target populations were medical specialists, mental healthcare providers, nurses and dentists.

### 2.4. Quality Assessment

The Jadad scale [17] was used for the quality assessment of the clinical trials included in the systematic review. The Newcastle-Ottawa Scale (NOS) [18] was used to evaluate observational studies. 

## 3. Results

After duplicates were eliminated, 26 potentially relevant studies were identified. Following the screening of titles and abstracts, 15 studies were excluded as they did not meet the inclusion criteria. Ultimately, eleven articles were included in the review. Among these, seven were clinical trials and four were observational studies. Table 1 and Table 2 summarize the characteristics of the included studies. Three articles focused on the nurse population, six articles on medical specialists and two articles on dentists. Ten studies were conducted in the USA and just one was conducted in China.

The quality assessment of clinical trials presented scores between a minimum of 1/5 and a maximum of 5/5. The average score was 1/5, which indicates an overall low quality of the studies. These data might be related to the difficulty in applying the parameters of this evaluation scale to these type of interventions (for instance, the difficulty of administering a yoga session to a blind population). The quality assessment for observational studies gave a minimum score of 5/9 and a maximum score of 8/9; the recorded average score of 6.5/9 indicates an overall good quality of the included observational studies.

Lastly, it can be stated that the studies included in our literature review addressed the same conclusions and were of average quality.

### 3.1. Physical Problems in Healthcare Professionals

Operating within a difficult setting, healthcare workers run into many stressful events that can compromise their state of health [22]. Often, as a consequence of the demand to maximize performance and productivity at work, health professionals’ needs to release emotional and physical tensions are ignored [23]. This inevitably results in serious issues affecting their health [24]. 

Work-related issues can be physical or affect the subject’s emotional network [22]. In the first situation the musculoskeletal apparatus is put under pressure. In the workplace there are situations that create a bio-mechanical overload, muscle tension and fatigue [25]. A stationary and inappropriate posture, the effects of repetitive tasks that require an accurate “execution” and that limit normal movement can be the cause of a physical injury [26]. Consequently, chronic degenerative diseases of the spine, neck and upper limbs affecting doctors, surgeons, dentists and nurses are frequent in the healthcare sector [27,28,29].

Dental hygienists represent a relevant example of this issue as they often experience neck and back pain, eye problems, muscle tension, headaches or carpal tunnel syndrome. These medical conditions are caused by poor posture, demanding schedules and long working hours, which can have a substantial negative impact on the physical well-being of the person and afflict the quality of professional life and job satisfaction [8].

Likewise, nurses are often exposed to physical stress resulting from lifting and moving heavy weights, which can lead to physical injury as backpain or herniated disc and that can interfere with their professional life [20].

### 3.2. Psycho-Emotional Problems in Healthcare Professionals

Due to their exposure to high stress levels, healthcare professionals often experience severe anxiety conditions that may lead to psycho-emotional disruption and ultimately result in burnout [30]. The psychologist Freudenberger coined the term “burnout” to describe the symptoms of professional exhaustion [31]. This word indicates the psycho-physical depletion of healthcare providers who gradually lose the ability to adapt to daily stress within the workplace [32]. The subject presents a combination of anxiety and distress, and consequently becomes unstable and neurotic [33].

Maslach, Professor of psychology at the University of California, Berkeley is considered one of the most important scientists regarding the burnout phenomenon, which he defines as “an emotional depletion syndrome, a depersonalization, and a reduction in personal abilities” that may arise in subjects whose professions “are concerned with the people” [34].

The consequences of this condition of psycho-physical distress hinder the well-being of the workforce and affect patients’ health, with both financial and organizational implications for healthcare systems.

### 3.3. The Use of Yoga as a Tool for the Welfare of the Healthcare Worker

Starting from their university training, medical and nursing students are exposed to heavy workloads, excessive stress and a competitive atmosphere [35,36]. These factors can negatively affect their future professional lives [1,2]. For this reason, it is necessary to identify an effective approach offering targeted solutions for stress management and contributing to achieving a healthy condition for these professional categories [37].

The discipline of yoga was originally defined as confined within a predominantly spiritual and meditative field. It evolved over time and assumed a more multifaceted nature of mixed techniques that combine mental with physical well-being [38]. This integrated approach guarantees holistic benefits for the individual. Firstly, aerobic and anaerobic exercises improve the musculoskeletal structure, the insulin circuit, the hormone system, and the metabolism [39,40,41]. Furthermore, the benefits achieved from sports or gymnastic exercise are enhanced by the association with meditation activities and psychoanalytic techniques that help to induce relaxation and self-awareness; this technique helps to improve cognitive flexibility and increase attention, acceptance and control of emotional reactions [42,43].

It is recognized that the daily practice of simple exercises strengthens the body and helps individuals to manage the burdens of life, work and interpersonal relationships [44]. Until a couple of decades ago, the term “meditation” was absent from medical textbooks and scientific articles. However, this approach is now becoming widely acknowledged. The examined scientific literature dealt mainly with the possibility of using meditation techniques—yoga in particular—to manage physical and mental stress, and their use has recently been hypothesized as a possible answer to work-related issues of healthcare workers [38].

From the scientific literature review it was seen that health professionals are receptive to this matter and perceive demanding workloads, intense care provision, conflicting expectations, patient management and family expectations as stressful factors that inevitably negatively affect their own health [3]. These elements are especially highlighted by some professional categories such as palliative care clinicians who are exposed to severe stress in their particularly challenging area of work. As an alternative to conventional cognitive training, healthcare workers recognize mind–body skills as important tools that can help them in managing their work activities [3]. Nurses for example, the largest group of healthcare professionals, are exposed to both physical and psychological stressors. In an anonymous survey conducted by Kemper et al., 50% of respondents were shown to have great expectations in mind–body training aimed at reducing anxiety, gaining serenity and a greater psychic well-being. About 99% of respondents reported having tried a mind–body practice in the 12 months prior to the study and among these 34% declared participation in meditation activities such as yoga, chi or qigong [20]. Conversely, in the category of dental hygienists references to yoga were more focused on solving problems related to low backpain and neck pain [8]. 

Moreover, clinical trials are useful in establishing the evidence of a greater effectiveness of this innovative approach and can highlight the potential of such disciplines by comparing them to normal cognitive trainings. 

Fang and Li’s clinical trial examines a population of 120 nurses randomized into two groups: one group followed a yoga program, while the other group followed no program. A six-month follow-up analysis shows that the yoga group reports reductions in stress levels (χ2 = 16.449; *p* = 0.001) Change using “χ2” [6]. The study of Alexander et al., conducted on a nurse population involved in an eight-week yoga program, also shows higher levels of self-care compared to a population who did not participate [5]. Riley’s work, in addition to giving an estimate of how yoga can improve the physical and psychological health status of staff, estimates how yoga programs are more effective than cognitive training programs in determining a better mental well-being and a reduction of stress-related consequences [19]. Finally, Bond et al. carried out a trial focused on medical students that were involved in an 11-week mind–body course: Statistically significant improvements were recorded in self-regulation values, which rose from 3.49 to 3.58 (*p* = 0.003), and in self-compassion values, which arose from 2.88 to 3.25 (*p* = 0.04). The perceived stress scores decreased from 1.55 to 1.48, and empathy levels increased from 5.64 to 5.80, however these values were not statistically significant (*p* = 0.70 and *p* = 0.30, respectively) [1].

## 4. Discussion

Increasing the use of yoga and meditation can provide a valid help to the healthcare workers in achieving a stable psycho-physical well-being that enhances their value within their work environment. This is mainly through simple meditation exercises which do not require a specific environment and can even be performed in any workplace setting [45]. The yoga program discussed in the study of Klatt et al. emphasizes the advantages of an activity that prove to be feasible and adaptable to the working environment [15]. Furthermore, the implementation of an individualized approach could guarantee an effective impact on the specific health needs of the employees. Programs such as Mindfulness in Motion (MIM) are delivered within the workplace and structured in short sessions that do not impact professional routine [15]. Alternatively, these yoga sessions could be organized alongside other activities such as art, music and writing. Along these lines, the Arts-in-Medicine program (AIM) described by Repar and Patton was devised. In addition, in this case, improvements were seen in the perceived levels of stress and overall well-being of individuals [7].

Within the scope of occupational medicine there are many interventions that can improve the psychological and physical well-being of healthcare workers, such as rescheduling working shifts or frequency and duration of breaks [46]. Nevertheless, the studies in this systematic review have shown an evident effectiveness of the yoga approach and suggest that a synergistic positive effect could be achieved when combining it with the other above-mentioned interventions [47].

The weaknesses of this analysis are undeniably related to the low numerical consistency of studies present in the literature and at the same time to the heterogeneity of the interventions that results in a difficulty to make any comparison. The studies included in this review are relatively recent: 2007 is the earliest year of publication and this highlights even more the innovation and the broad potential of this insight.

Alternatively, the strengths of this study are represented by the revolution that this type of approach brings for the management of healthcare workers as the costs for implementing them are low in comparison to the considerable projected costs of financial and welfare benefits. Healthcare administrators, who are called upon to offer efficient solutions to workers’ health issues, must acknowledge the importance of this simple and effective approach.

## 5. Conclusions

According to the published literature, yoga is effective in the prevention and management of musculoskeletal and psychological issues. In addition to an improvement in physical problems and in quality of sleep, both stress levels and burnout are consistently reduced in subjects who practice yoga techniques and mind–body meditation. Although the data published highlight the full potential and possible benefits derived from these techniques, in order to warrant a widespread diffusion as a daily practice, it would be necessary to broaden the subject further and acquire more robust scientific evidence by designing and implementing research studies equipped with a solid methodological structure on bigger sample groups.

## Figures and Tables

**Figure 1 jcm-08-00284-f001:**
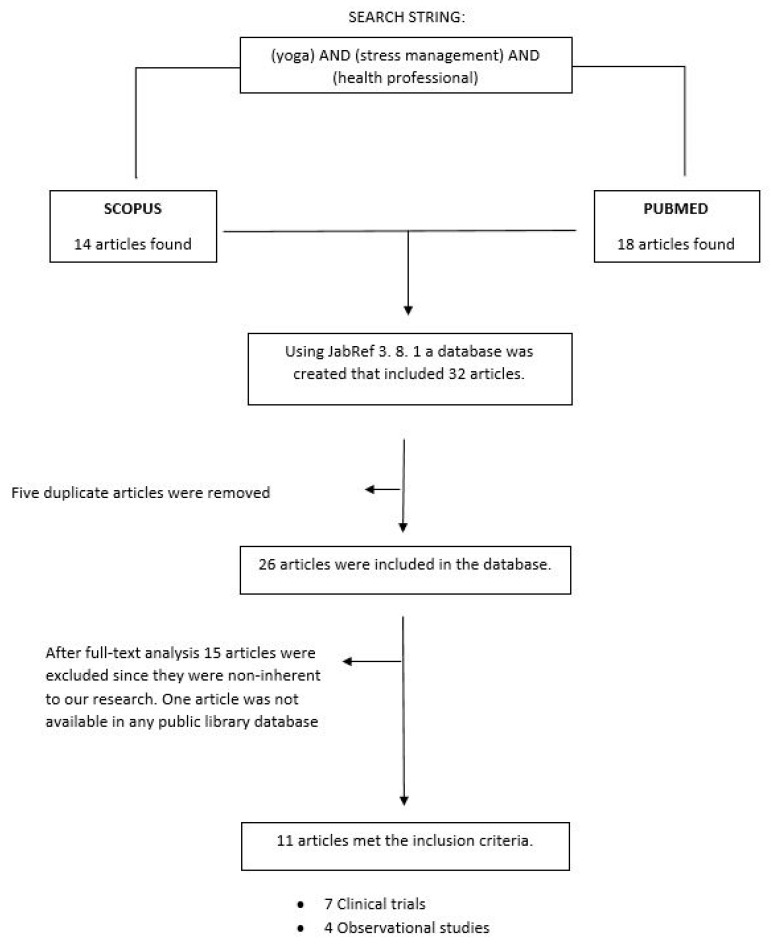
Research flowchart displaying selection and analysis of the items included in the systematic review.

**Table 1 jcm-08-00284-t001:** Clinical trials included in the systematic review.

Authors	Country	Year	Results	Jadad Score
Alexander et al. [5]	USA	2015	Yoga should be used to improve physical and mental health, to promote self-care and preventing burnout among nurses. Yoga intervention in this clinical trial lasted 8 weeks and was intended to develop self-reflection, self-care and self-discovery. Mind–body practices are useful in managing stress and building emotional resilience. To evaluate lifestyles, the authors used Health Promoting Lifestyle Profile II (HPLP-II); curiosity, acceptance and feasibility were evaluated with the Freiburg Mindfulness Inventory (FMI) while burnout was measured with the Maslach Burnout Inventory (MBI).	1
Bond et al. [1]	USA	2013	Medical students were evaluated for Jefferson’s Scale of Physician Empathy, Cohen’s Perceived Stress Scale, Self-Regulation Questionnaire and Self-Compassion Scale. The intervention consisted of an 11-week program of yoga and meditation and of a post-intervention evaluation. Statistically significant improvements in self-regulation and self-compassion of students were recorded. Changes in empathy and perceived stress were also recorded, although these values did not reach statistical significance.	3
Fang and Li [6]	China	2015	Yoga techniques were offered to nurses to improve sleep quality and reduce stress after work shifts. The group who received the intervention followed the yoga sessions twice a week (50/60 min per session) and demonstrated better sleep quality and reduced work stress. Yoga can improve back pain and quality of life. Sleep quality was evaluated using the Pittsburgh Quality Index (C-PSQI) while for work pressure, the authors used the Questionnaire on Medical Worker’s Stress (QMWS).	5
Klatt et al. [15]	USA	2015	This clinical trial was structured on an 8-week yoga program associated with a day-to-day work of 20 min of meditative awareness. The program was called MIM (Mindfulness In Motion) and consisted of a less time-invasive method to be administered in the work environment, based on meditation awareness, yoga stretching, relaxing music, and daily individual meditation practice. Resilience was measured with the Connor-Davidson Resiliency Scale (CD-RISC). The work commitment was evaluated through the Utrecht Work Engagement Scale (UWES). This study was designed for stressed workers, such as health care professionals of intensive care units.	1
Repar and Patton [7]	USA	2007	This Arts In Medicine (AIM) Program aimed to resolve compassion, fatigue, and chronic pain among nurses. Massages, poetry, listening to live music and visual arts, in combination, helped to improve the quality of life.	1
Riley et al. [19]	USA	2017	This large randomized clinical trial included two studies to compare the impact of Cognitive Behavioral Stress Management (CBSM) and Yoga-Based Stress Management (YBSM). Stress was associated with high levels of blood pressure, weight gain, anger, depression, anxiety, and reduced quality of life with worse health behaviors (diet, exercise, alcohol use). This trial, which included an 8-week yoga program, also focused on the consequences of stress, such as suicide and costs associated with tournaments and absenteeism. Among the questionnaires included in the study there were Depression, Anxiety and Stress Scale (DASS-21), SF12 and Self-Compassion Scale-Short Form (SCS).	2
Shirey [2]	USA	2007	The goal of this study was to investigate evidence-based solutions to reduce anger and stress. Consciousness-based interventions have shown a reduction in stress. The instrument used to evaluate empathy, considering mood changes, was the Interpersonal Reactivity Index (IRI).	1

**Table 2 jcm-08-00284-t002:** Observational studies included in the systematic review.

Authors	Country	Year	Results	NOS
Chismark et al. [8]	USA	2011	This survey conducted on dental hygienists concerned the use of complementary medicine, including yoga and meditation to manage muscle-skeletal pain and obtain a better overall health and career satisfaction.	8
Kemper et al. [20]	USA	2011	This study reported the results of an online survey administered to nurses to evaluate their experience with meditation, prayer and mind–body practices. Expectations about the benefits of physical, emotional, mental and spiritual health and preferences on the type and structure of meditation training were also investigated.	5
Perez et al. [3]	USA	2015	The target population of this study was represented by palliative care clinicians. Factors of interest in the interviewed population were the challenges of managing a heavy workload (establishing skills and recognizing limitations); patient-related factors (patient management and case intensity); emotional and professional limits (limited resources, conflicting needs and expectations).	5
Sulenses et al. [21]	USA	2015	This survey was structured in 22 questions to evaluate the commitment to yoga practice, barriers to yoga practice, acceptability as additional treatment for physical and mental health and the characteristics of the participants. The aim was to demonstrate the under-utilization of yoga as a complementary alternative medicine resource (CAM).	8

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
