# Peer review of "The Use of Yoga to Manage Stress and Burnout in Healthcare Workers: A Systematic Review"

_jcm, 2019, doi:10.3390/jcm8030284_

Round 1
Reviewer 1 Report
This systematic review addresses an important question about the use of yoga for stress in healthcare workers. The manuscript to PRISMA reporting standards. The paper needs additional citations to support claims throughout (probably another 30 citations are warranted). Citations need to be recent and relevant too- (e.g., not a quotation about burnout from 1974). The paper could benefit from rewording (wording throughout is awkward). The paper would benefit from reorganizing paragraphs for greater clarity too, as they aren’t grouped by topic area and don’t flow from one paragraph to the next (e.g., define burnout clearly in introduction and expand on how this research moves the field forward in the discussion). For example:
Introduction:
1. Recommend eliminating the word syndrome after burnout.
2. Need additional literature support for the following points:
a. This pattern has captured the attention of healthcare managers who are now seeking solutions to these problems.
b. Burnout and stress may produce dissatisfaction and anxiety in the workplace, a reduced quality of care and a higher risk for patients.
3. A better description of why stress and burnout are important to address in the first paragraph would be nice. It’s not all about empathy towards patients (although that is part of it).
4. Aims are clear.
Materials and methods:
5. Search strategy well described- would have benefitted from the inclusion of terms related to healthcare professional too, but the study is completed at this point…
6. Nice flowchart.
7. Seems like mental healthcare providers are targeted too- might be good to mention this as well.
Results and Discussion:
8. I enjoyed that you included quality ratings.
9. Nice summary tables.
10. Paragraphs are not grouped by topic, and do not flow well from one paragraph to the next. Reorganization needed.
11. Wording in some places is awkward.
12. Citations are desperately needed throughout the entire paper. After almost every sentence. For example:
a. Operating in a healthcare setting, workers encounter numerous stressful events that can 2 compromise their state of health.
b. Often in the workplace, because of the demand to maximize performance and productivity… 3
c. This inevitably results in serious issues affecting their health.
d. Even during their training, medical and nursing students are exposed to heavy workloads, 6 excessive stress and a competitive atmosphere
e. Work-related issues can be of a physical nature or affect the subject's emotional sphere.
f. In the first, the musculoskeletal apparatus is put under stress.
g. In the workplace, there are situations that create a bio-mechanical overload, muscle tension and fatigue.
h. A stationary and inappropriate posture, the effects of repetitive tasks that require an accurate "execution" and that limit normal movement can be the cause of physical injury. Chronic-degenerative diseases of the spine, neck and upper limbs affecting doctors, surgeons, dentists and nurses are frequent in the healthcare sector.
13. The Freudenberger reference is very outdated. Referencing research from the past ten years is recommended: The subject presents a combination of anxiety and distress and, consequently, becomes unstable and neurotic
14. Citations are incorrectly numbered in the text- do not match up with references.
Thanks for your work in this area.
Author Response
Dear Editorial Board,
By this document we want to highlight all the changes implemented in our study.
Reviewer 1
Comments and Suggestions for Authors
This systematic review addresses an important question about the use of yoga for stress in healthcare workers. The manuscript to PRISMA reporting standards. The paper needs additional citations to support claims throughout (probably another 30 citations are warranted). Citations need to be recent and relevant too- (e.g., not a quotation about burnout from 1974). The paper could benefit from rewording (wording throughout is awkward). The paper would benefit from reorganizing paragraphs for greater clarity too, as they aren’t grouped by topic area and don’t flow from one paragraph to the next (e.g., define burnout clearly in introduction and expand on how this research moves the field forward in the discussion). For example:
According to the suggestions from Reviewer 1 our study was implemented supporting claims with further citations: added references are relevant and recent (last 10 years). The paper has received language revision and has been reorganized in the structure.
Introduction:
1. Recommend eliminating the word syndrome after burnout.
The word “syndrome” has been removed after burnout.
2. Need additional literature support for the following points:
a. This pattern has captured the attention of healthcare managers who are now seeking solutions to these problems.
b. Burnout and stress may produce dissatisfaction and anxiety in the workplace, a reduced quality of care and a higher risk for patients.
Relevant and recent references have been added to support the sentences through the text.
2. A better description of why stress and burnout are important to address in the first paragraph would be nice. It’s not all about empathy towards patients (although that is part of it).
The introduction paragraph has been implemented with a concise description of the importance and effects of stress and burnout.
4. Aims are clear.
Materials and methods:
5. Search strategy well described- would have benefitted from the inclusion of terms related to healthcare professional too, but the study is completed at this point…
6. Nice flowchart.
7. Seems like mental healthcare providers are targeted too- might be good to mention this as well.
We have accepted the comment and implemented the text adding a referral to mental health providers.
Results and Discussion:
8. I enjoyed that you included quality ratings.
9. Nice summary tables.
10. Paragraphs are not grouped by topic, and do not flow well from one paragraph to the next. Reorganization needed.
According to the reviewer’s suggestion we have reorganized the results’ paragraph highlighting the topic of physical and psychological work-related issues.
11. Wording in some places is awkward.
We have performed language revision through all the text.
12. Citations are desperately needed throughout the entire paper. After almost every sentence. For example:
a. Operating in a healthcare setting, workers encounter numerous stressful events that can 2 compromise their state of health.
b. Often in the workplace, because of the demand to maximize performance and productivity… 3
c. This inevitably results in serious issues affecting their health.
d. Even during their training, medical and nursing students are exposed to heavy workloads, 6 excessive stress and a competitive atmosphere
e. Work-related issues can be of a physical nature or affect the subject's emotional sphere.
f. In the first, the musculoskeletal apparatus is put under stress.
g. In the workplace, there are situations that create a bio-mechanical overload, muscle tension and fatigue.
h. A stationary and inappropriate posture, the effects of repetitive tasks that require an accurate "execution" and that limit normal movement can be the cause of physical injury. Chronic-degenerative diseases of the spine, neck and upper limbs affecting doctors, surgeons, dentists and nurses are frequent in the healthcare sector.
We added citations and references through all the text, considering their relevance and the year of publication (only last 10 years studies were considered).
13. The Freudenberger reference is very outdated. Referencing research from the past ten years is recommended: The subject presents a combination of anxiety and distress and, consequently, becomes unstable and neurotic
We agree with the reviewer that the Fraudenberger citation from 1974 was very outdated but we believed that it could be interesting to furnish a historic referral to the first utilization of the word “burnout” in the scientific literature. However, we implemented this paragraph with more recent citations.
14. Citations are incorrectly numbered in the text- do not match up with references.
We reviewed all the document references and citations and we rectified all the inconsistencies.
Reviewer 2 Report
This is an interesting paper on work-related health management techniques. Indeed there is a need to act on regulating stress among the healthcare professionals at the workplace and yoga may be considered.
However, in the Discussion section there is need to mention other approaches such as rescheduling working routines, frequency and duration of breaks, availability of other exercise opportunities etc and provide some comparison or evidence of synergistic effect with yoga programmes.
Moreover, some further explanation on the low quality of the cinical trials should be offered to the reader.
Another element that needs mention is the degree of feasibility to introduce yoga programmes in many working environments as well as the individualised approach that may be required to increase the beneficial effects on the employees.
Author Response
Dear Editorial Board,
By this document we want to highlight all the changes implemented in our study.
Reviewer 2
Comments and Suggestions for Authors
This is an interesting paper on work-related health management techniques. Indeed there is a need to act on regulating stress among the healthcare professionals at the workplace and yoga may be considered.
However, in the Discussion section there is need to mention other approaches such as rescheduling working routines, frequency and duration of breaks, availability of other exercise opportunities etc and provide some comparison or evidence of synergistic effect with yoga programmes.
We accepted the reviewer’s suggestions and implemented the text with the suggested changes.
Moreover, some further explanation on the low quality of the cinical trials should be offered to the reader.
We accepted the reviewer’s suggestion and implemented the text giving a concise explanation of the reason for the low quality score of clinical trials.
Another element that needs mention is the degree of feasibility to introduce yoga programmes in many working environments as well as the individualised approach that may be required to increase the beneficial effects on the employees.
We accepted the reviewer’s suggestions and implemented the text with the suggested changes.
Round 2
Reviewer 1 Report
The authors responded well to suggested edits.